# Eruption Pattern of Third Molars in Orthodontic Patients Treated with First Permanent Molar Extraction: A Longitudinal Retrospective Evaluation

**DOI:** 10.3390/jcm12031060

**Published:** 2023-01-30

**Authors:** Lisa J. Langer, Nikolaos Pandis, Maria R. Mang de la Rosa, Paul-Georg Jost-Brinkmann, Theodosia N. Bartzela

**Affiliations:** 1Department of Orthodontics and Dentofacial Orthopedics, Charité–Universitätsmedizin Berlin, Corporate Member of Freie Universität Berlin and Humboldt-Universität zu Berlin, 14197 Berlin, Germany; 2Department of Orthodontics and Dentofacial Orthopedics, Dental School/Medical Faculty, University of Bern, 3012 Bern, Switzerland; 3Department of Orthodontics, Faculty of Medicine Carl Gustav Carus, Technische Universität Dresden, 01307 Dresden, Germany

**Keywords:** first permanent molars, extraction therapy, orthodontics, third molars, eruption prognosis

## Abstract

The aim of this study was to evaluate angular and positional changes in the second (M2) and third molars (M3) of orthodontically treated patients undergoing a first molar (M1) extraction. A retrospective longitudinal study with a sample of 152 pre- and post-treatment panoramic radiographs was conducted. Thirty-nine patients (51.3%) were orthodontically treated with M1 extraction and thirty-seven (48.7%) were treated without extraction. Angulations of M2 and M3 relative to the infraorbital (IOP) and the palatal planes (PP) were measured and compared between the groups before orthodontic treatment (T1) and after the completion of orthodontic space closure (T2). The prognosis of M3 eruptions was evaluated by assessing their horizontal and vertical position (inclination) using different classification systems. The angular (*p* < 0.001) and inclination improvement (*p* < 0.01) of the maxillary M3 was significant for the M1 extraction group. The mandibular M3 inclination significantly improved (*p* < 0.01), whereas the groups’ angulation and vertical position were not significantly different. These findings suggest that extraction therapy has a favorable effect on the maxillary M2 and M3 angulation, but not on the mandibular. M1 extraction showed a signi- ficant effect on the horizontal position of M3 and thus may improve the eruption space and prognosis.

## 1. Introduction

Treating young patients with deep carious lesions in permanent first molars (M1) or molar-incisor hypomineralization (MIH) is challenging for dentists and orthodontists alike. The prevalence of MIH has risen dramatically over the recent years, up to almost 40% in 6–14-year-olds [1,2], due to the global increased dentists’ awareness. Despite the more frequent MIH occurrence in everyday clinical practice, few studies have addressed the problem from an orthodontic point of view [3]. Early diagnosis of MIH is essential to ensure adequate management of the affected teeth. If MIH diagnosis occurs late, an invasive treatment approach, such as extraction, may be required. This has been further aggravated by the pandemic situation, which has caused delays in diagnosis and dental treatment [4].

The extraction of severely decayed M1 in combination with orthodontic space closure may influence the eruption prognosis of the third molars (M3).

The M3 are the most frequently impacted teeth [5,6], with an inconsistent range between 3% and 57% worldwide, influenced by the method and diagnostic criteria applied [7]. Risks associated with partially-unerupted or impacted M3 are periodontitis, other inflammatory conditions, follicular cysts, benign neoplastic disease, and resorption of the M2 roots [5]. Nevertheless, prophylactic removal of asymptomatic disease-free M3 is not generally recommended, due to the risk of surgical intervention and cost-effectiveness [8]. In patients with congenitally missing teeth, loss of M1 or M2 due to caries or orthodontic extraction planning, removal of the M3 should be deferred [5]. Moreover, the premature loss of M1 has an accelerating effect on the development and the eruption of M3 [9,10] in both jaws [9,10,11,12,13,14]. The M3 impaction incidence might be lower after orthodontic extraction treatment. Previous studies found a greater impact on maxillary M3 than on mandibular M3 after M1 orthodontic extraction [9,12]. Other factors, such as decreased retromolar space, severely malpositioned M3, or extraction time of the M1 may also impact the M3 eruption prognosis [10,14,15,16,17]. Craniofacial parameters, including steepness of the gonial angle [18], and mandibular and facial growth patterns [19], may contribute to the M3 eruption pattern. Nevertheless, it remains unclear to what extent an orthodontic space closure following M1 extraction influences the eruption prognosis and final position of the M3.

Therefore, the aim of this study was to assess the angulation and position changes of M2 and M3 and the eruption prognosis of M3 after M1 extraction orthodontic treatment.

## 2. Materials and Methods

This longitudinal retrospective study was conducted according to the guidelines of the Declaration of Helsinki and approved by the Ethics Committee of Charité–Universitätsmedizin Berlin, Germany (EA2/231/18).

Digitized panoramic radiographs of 76 growing individuals taken before (T1) and after orthodontic space closure (T2) were included. All patients were recruited from the Department of Orthodontics and Dentofacial Orthopedics, Charité–Universitätsmedizin Berlin, and two private orthodontic practices in Berlin, Germany. A sample of 39 patients (24 females, 15 males, and a mean age of 13.5 ± 1.8 years) was treated with at least one M1 extraction. In the maxilla, 17 patients were treated with bilateral M1 extraction (*n* = 34) and 12 with unilateral (*n* = 12). In the mandible, 19 patients received bilateral extraction treatment of M1 (*n* = 38) and 12 unilateral (*n* = 12). The control group consisted of 37 patients (23 females, 14 males, and a mean age of 14.5 ± 4.9 years) treated without extraction. Both groups were treated with multibracket appliances. The M2 erupted in the oral cavity as expected at the patients’ age at T1. All included patients were of Caucasian ethnic background. Patients with congenitally missing teeth, craniofacial anomalies or obvious asymmetries that were clinically or radiographically recognizable, syndromes, or treated with distalizing devices were excluded from this study. The orthodontic diagnosis and planning were collected from patients´ pretreatment lateral cephalograms and treatment records. For determining the longitudinal axis of M3, at least a fully mineralized crown formation (stage 4, according to Demirjian’s classification [20]) at T1 was required. The infraorbital (IOP) and the palatal (PP) planes were the references selected for the M2 and M3 angulation measurements. The IOP was constructed by the most inferior points of the left and right infraorbital margins. The PP was defined by the most cranial points of the curvature of the palate (Figure 1). The angles formed between these reference planes and the longitudinal axes of M2 (M2/IOP, M2/PP) and M3 (M3/IOP, M3/PP) (Figure 1) were measured and compared at T1 and T2.

The reference planes were traced and angles were measured on the digital radiographs with Sidexis XG software (version 2.63, Dentsply Sirona, Bensheim, Germany).

Moreover, the prediction of M3 eruption was evaluated by Archer’s classifications [21,22,23] in the maxilla (horizontal and vertical) (Figure 2 and Figure 3) and by Winter’s (Figure 4) [21,24] and Pell & Gregory’s classification (Figure 5) [21,25,26] in the mandible. We propose merging the subgroups of the classification systems used as good, questionable, and bad according to the prediction of spontaneous M3 eruption (Figure 2, Figure 3, Figure 4 and Figure 5). The first author (L.L.) made all measurements. Two calibrated authors (L.L. and M.M.) measured thirty randomly selected panoramic radiographs for the inter-rater agreement. The first author (L.L.) evaluated twenty patients (forty panoramic radiographs) twice, with at least a one-month interval, for intra-observer reliability. Intraclass correlation coefficient (ICC) with a two-way mixed effects model [27] was calculated for the angular measurements (M2/IOP, M3/IOP, M2/PP, M3/PP) and Cohen’s kappa coefficient (κ) [28] for the categorical measurements (Demirjian’s, Archer’s, Winter’s, and Pell & Gregory’s classifications). Descriptive statistics were calculated for baseline characteristics in the extraction and non-extraction groups. The angular measurements were compared using analysis of covariance (ANCOVA), where the effect of treatment at T2 was estimated after adjusting for the baseline (T1) angular measurement. A random effect (random intercept) ordinal logistic regression was used to assess the adjusted impact of extraction/non-extraction on the final position of M3 according to the different classification systems (Figure 2, Figure 3, Figure 4 and Figure 5).

The statistical significance was set at *p* < 0.05. All analyses were conducted in Stata 16 (Stata Corp, College Station, TX, USA).

## 3. Results

For the angular measurements, the intra-observer reliability ICC was 0.99 with a 95% confidence interval (95% CI) = 0.997–0.999. For the inter-observer reliability, ICC ranged between 0.98 and 0.99, with 95% CI = 0.96–0.99, demonstrating excellent reliability [27]. For the categorical measures, kappa ranged between 0.86 and 1.00 for all measurements, demonstrating an almost perfect strength of agreement [28].

The descriptive data for both groups are presented in Table 1. The orthodontic treatment time was significantly longer for patients treated with extraction of M1 than those treated without extraction (*p* < 0.001), as expected. The period between the pre- and post-treatment radiographs was 3.8 ± 1.3 years in the non-extraction group and 3.7 ± 1.4 years in the extraction group.

Table 2 shows the baseline adjusted estimates of the effect of treatment (extraction/non-extraction) on each of the variables (M2/IOP, M2/PP, M3/IOP, M3/PP) separately per jaw. On average, the maxillary M2/IOP, M2/PP, M3/IOP, and M3/PP were −8.72, −8.83, −10.17, and −10.04 units lower in the extraction group compared to the non-extraction group, respectively. These results were all statistically highly significant (*p* < 0.001). In the mandible, the differences between extraction/non-extraction were smaller. All average angular measurements of the extraction and non-extraction groups at the different time points (T1, T2) are presented in Figure 6.

In the maxilla, the positional changes of M3 were evaluated using Archer’s classifications (Figure 2 and Figure 3) [23]. The inclination of maxillary M3 relative to the longitudinal axis of adjacent M2 (Figure 2) improved significantly (*p* = 0.005). However, the vertical positional changes (Figure 3) were non-significant between the extraction and non-extraction groups at the evaluated time points (*p* = 0.07). In the mandible, the classifications by Winter (Figure 4) [24] and Pell & Gregory (Figure 5) [25] were applied to analyze the positional changes of M3. Regarding Winter’s classification, changes in the mandibular M3 inclination compared to the long axis of M2 showed significant improvement, mainly in the M1 extraction group (*p* = 0.002). The position changes of mandibular M3, according to Pell & Gregory’s horizontal classification (PGH), were significant in the extraction group at T2 (*p* < 0.001). However, the position of M3, according to Pell & Gregory’s vertical classification (PGV), showed no significant change between the groups (*p* = 0.14). The eruption prognosis of M3 in the two groups and time points according to the different classification systems is presented in Figure 7.

The M3 development analysis based on Demirjian’s classification [20] did not differ significantly between the extraction and non-extraction groups at T1 (*p* = 0.48) and T2 (*p* = 0.12).

We were able to follow up with 59% of the patients in the extraction group and 94.6% of the non-extraction group for 23 ± 13.1 months. The end-point evaluation was the eruption or the final eruption prognosis of the M3. Of those patients treated with M1 extraction, 65.2% showed a spontaneous eruption of the M3. In 21.7% of the patients, the position of M3 was improved after orthodontic treatment, and in 13.1% of patients, M3 surgical exposure was required. All M3 impactions of the extraction group were observed in the mandible. In only 5.7% of the non-extraction patients, spontaneous eruption of M3 occurred. Most non-extraction treated patients (94.3%) were referred for surgical extraction of M3.

The random effects analysis (Table 3) shows that only the extraction was a significant predictor (odds ratio (OR) = 1.96; 95% CI: 1.04–3.72; *p* = 0.04) for the final position (inclination) of maxillary M3 (Archer’s horizontal classification, Figure 2), whereas time point, sex, and dental crowding were not. In the mandible, time was a significant predictor (*p* < 0.001) for the inclination (Winter’s classification, Figure 3) and vertical position (PGV, Figure 4) of M3. Both M1 extraction (OR = 2234.22; 95% CI: 11.35–439,962.1; *p* = 0.004) and time (OR = 5459.58; 95% CI: 16.53–1,802,824; *p* = 0.004) were highly associated with the horizontal positional changes of M3 (PGH, Figure 4). Severe dental crowding (>5 mm) was negatively associated only with the changes in the inclination of mandibular M3 (OR = 0.004; 95% CI: 0.00002–0.95; *p* = 0.048). No significant difference was found between female and male patients. However, the previous estimates cannot be considered reliable given the OR values and the corresponding 95% CI.

## 4. Discussion

The dilemma of extraction or extensive restorations of M1, affected by MIH, or deep carious lesions, should be interdisciplinarily evaluated. Early diagnosis of MIH is essential for the prognosis of affected teeth to allow the provision of preventive or restorative intervention [3,29]. The long-term eruption prognosis of M2 and M3 and the associated risk factors in the case of M1 extraction therapy are the most critical factors to be considered. In severe MIH, not only M1 but also M2 may be affected, making treatment decisions more challenging [30]. Complications such as sensitivity and discomfort and restoration failures could be encountered. A health economics analysis stated that the removal of compromised M1 is not generally more cost-effective than preservation [31].

Our study showed that M1 extraction has a favorable effect on maxillary M2 and M3 eruption patterns. In contrast, mandibular M2 angulation changes during treatment were non-significant between the extraction and non-extraction groups. The M3 showed greater uprighting in the M1 extraction group. However, all mandibular measures at T2 showed no significant difference between the groups. A reason for that might be their severe mesial inclination [15,16,32].

Moreover, the timing of M1 extraction should be considered cautiously. Extraction of M1 before the eruption of M2 can result in spontaneous space closure [33], which is more effective in the maxilla than in the mandible [34,35,36]. Early extraction of mandibular M1 leads to uncontrolled tipping of the adjacent molars [14]. The recommended extraction time for mandibular M1 is between 8 and 11.5 years of age [37].

Furthermore, factors such as the angulation of M2 and the presence of M3 are important for M1 extraction planning. Nevertheless, a confirmed diagnosis of M3 agenesis can only be verified radiographically in some patients no younger than 14 years of age [38]. A study that assessed the orthodontists’ and oral surgeons’ prognostic prediction of M3 eruption concluded that in many patients, M3 erupted spontaneously, despite the initial recommendation for extraction [39]. Therefore, a longitudinal assessment of the eruption path should be considered [36] and not a single radiographic evaluation. Compensation extraction on the quadrants with sound molars is recommended for symmetric dental arches and balanced occlusion [40]. Unilateral extraction requires careful management of orthodontic anchorage to prevent a dental midline shift towards the extraction side [41]. Nevertheless, M1 asymmetric, compared to M1 symmetric extraction orthodontic treatment, has less impact on soft tissue and facial profile [42].

This study did not consider early extraction, dental arch asymmetry, and incisor proclination at the evaluated time points (T1 and T2). Severe arch length discrepancy and incisor protrusion are contraindications for early extraction of M1, and controlled M2 migration should be considered [43]. A study on twins showed an association between a small retromolar space or a constricted mandibular dental arch with an unfavorable prognosis for M3 eruption [44]. Future research should include these parameters.

Our results confirm previous studies [9,12,13] and showed an improvement in maxillary M3 inclination but no statistically significant difference in the angulation of mandibular M3 after M1 extraction.

Only angular measurements and categorical impaction classification systems have been used in this study. Therefore, the distortion and magnification factors of the orthopantomograms [45] did not affect the interpretation of the results [46]. No cephalograms were used for the retromolar space evaluation because of the overlapping anatomical structures and the magnification differences between the participating orthodontic practices [47,48].

The present study presented all angles relative to IOP and PP, as described in previous studies [49,50,51].

The horizontal planes (IOP and PP) can be easily traced based on skeletal structures on panoramic radiographs. These planes are more reliable [52] than the occlusal plane, which undergoes growth and orthodontic treatment modifications [53,54].

In order to increase measurement validity, both planes have been used in this study. Only panoramic radiographs and lateral cephalograms were available for the study sample. These images do not allow the angulations of M2 and M3 to be assessed in all planes, mainly if teeth are more buccally or lingually oriented. Future studies should use three-dimensional x-ray images, if available, for a more detailed evaluation. Still, these radiographs are not commonly performed due to increased radiation exposure.

Despite the retrospective character of the study, we were able to ensure a homogeneous distribution of the sample. A slight increase in patients with a vertical growth pattern in the extraction group (Table 1) was observed. This may have been an additional factor in the extraction treatment approach. Former studies found that patients with a dolichofacial growth pattern showed a twofold prevalence of mandibular M3 impaction than patients with a horizontal growth pattern (brachyfacial) [19,55,56].

No significant difference was found in patients´ age at T1 and T2 between the groups. Nevertheless, the individual radiographs were taken at different time points, and the treatment duration was longer in the extraction group. The increased treatment time in the extraction group may potentially influence the development stage and, consequently, the vertical position of M3. Nevertheless, Archer’s vertical classification for maxillary M3 (Figure 3 and Figure 6) and Pell & Gregory’s vertical classification for mandibular M3 (Figure 5 and Figure 6) did not differ significantly between extraction and non-extraction groups (*p* > 0.05). Furthermore, no significant difference was found between the groups at T2 (*p* = 0.12) in Demirjian’s classification. Thus, it can be assumed that the time difference when radiographs were taken is clinically irrelevant concerning the vertical development of M3. This can be related to our sample’s short observation time (T1–T2). We elaborated on a simplification of the classification systems (Archer, Winter, Pell & Gregory) by merging the subgroups for clarity, focusing mainly on the prognostic eruption of the M3.

Time and M1 extraction were relevant factors for horizontal positional changes of mandibular M3 (Pell & Gregory horizontal (Figure 4), Table 3). Around puberty, retromolar space increases, averaging 1.38 mm per year in non-treated orthodontic patients [57]. To what extent the changes in the examined patients were related to growth or the extraction of M1 cannot be clarified conclusively. However, there was a significant difference between extraction and non-extraction groups regarding the horizontal position of M3 in the mandible at T2 (Figure 7).

We could only follow up with 59% of the patients with M1 extraction therapy over a period of 24 ± 10.3 months. Spontaneous eruption of the M3 was observed in 65.2% of these patients. The mean age of these patients at T2 was 16.2 years (range 14.4–18.0) (Table 1). Due to the patients´ young age at T2, it is unclear whether the remaining M3 erupted later. Nevertheless, this result shows a tendency towards a favored eruption or eruption prognosis of M3 after M1 extraction, particularly in the extraction group. Further studies with a longer follow-up period may elucidate and maximize an accurate prediction of the M3 eruption pathway after orthodontic treatment.

M2 angular changes should be considered with the knowledge of possible optimization/uprighting by the multibracket appliance. Tipping of M2 due to orthodontic space closure [41,58] was also found in some investigated patients. Still, it was not examined in more detail whether this influenced the subsequent position of M3.

The findings of this study might help clinicians establish a more patient-centric treatment approach for patients with compromised M1.

## 5. Conclusions

The M1 extraction positively influenced the angulation of maxillary M3, whereas angulation of the mandibular M3 barely improved.

The extraction of M1 showed a positive effect on the position of M3 and M2 in both jaws in the horizontal plane, which may increase the retromolar space. Parameters such as sex and mild dental crowding had no impact on the final position of M3.

Patients with compromised M1 require a well-coordinated, individualized treatment approach. Long-term prognostic factors for the eruption of M3, possible complications associated with M1 extraction, or M3 orthodontic uprighting and surgical exposure should be considered. These findings should be considered with regard to the study’s retrospective character.

## Figures and Tables

**Figure 1 jcm-12-01060-f001:**
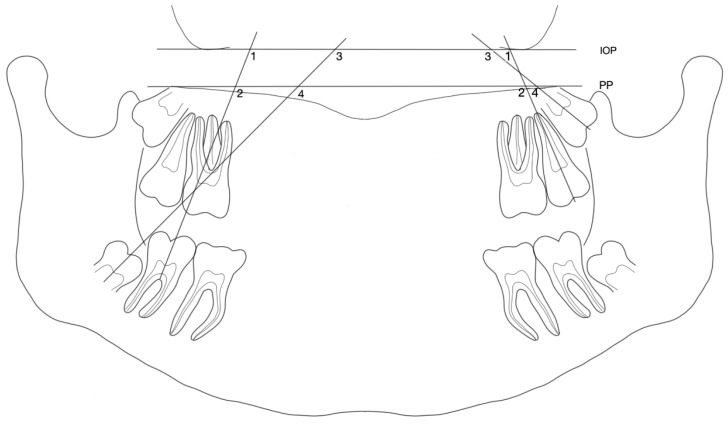
Diagram of angulation measurements. **1** mandibular/maxillary (Mb/Mx) second molar (M2) angulation to the infraorbital plane (IOP): **M2/IOP**; **2** Mb/Mx M2 angulation to the palatal plane (PP): **M2/PP**; **3** Mb/Mx third molar (M3) angulation to IOP: **M3/IOP**; **4** Mb/Mx M3 angulation to PP: **M3/PP**.

**Figure 2 jcm-12-01060-f002:**
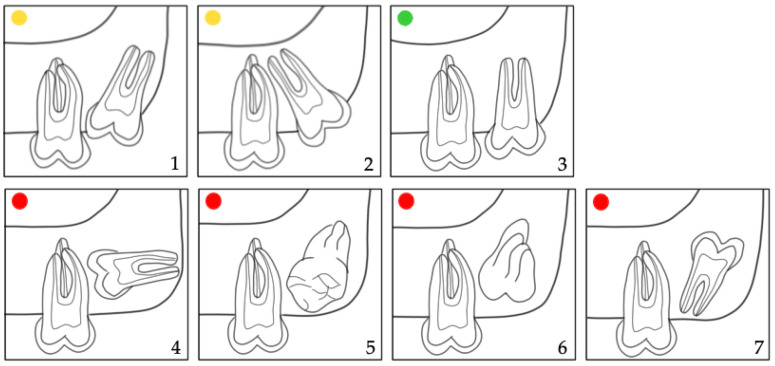
Archer’s (horizontal) classification for the inclination of the maxillary M3 relative to the longitudinal axis of M2 [23]. Subdivision according to the eruption prognosis of M3 in **1**,**2**: questionable (
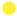
); **3**: good (
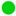
); **4**–**7**: bad prognosis (
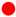
).

**Figure 3 jcm-12-01060-f003:**
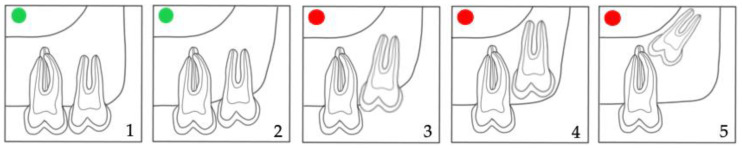
Archer’s (vertical) classification for the impaction depth of the maxillary M3 relative to the adjacent M2 [23]. Subdivisions according to the eruption prognosis of M3 in **1**,**2**: good (
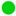
); **3**–**5**: bad prognosis (
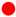
).

**Figure 4 jcm-12-01060-f004:**
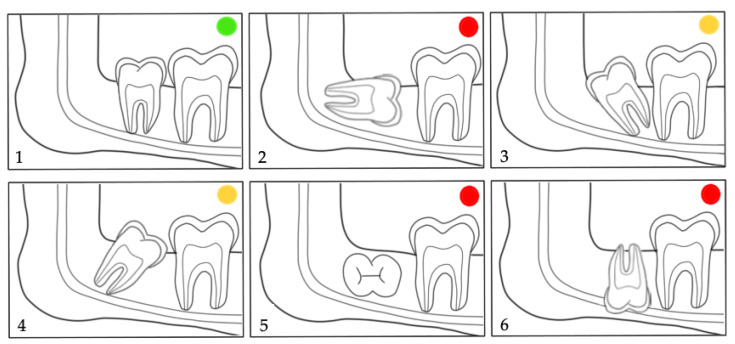
Winter’s classification for the inclination of the mandibular M3 relative to the longitudinal axis of M2 [24]. Subdivisions according to the eruption prognosis of M3 in **1**: good (
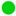
); **3**,**4**: questionable (
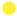
); **2**,**5**,**6**: bad prognosis (
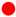
).

**Figure 5 jcm-12-01060-f005:**
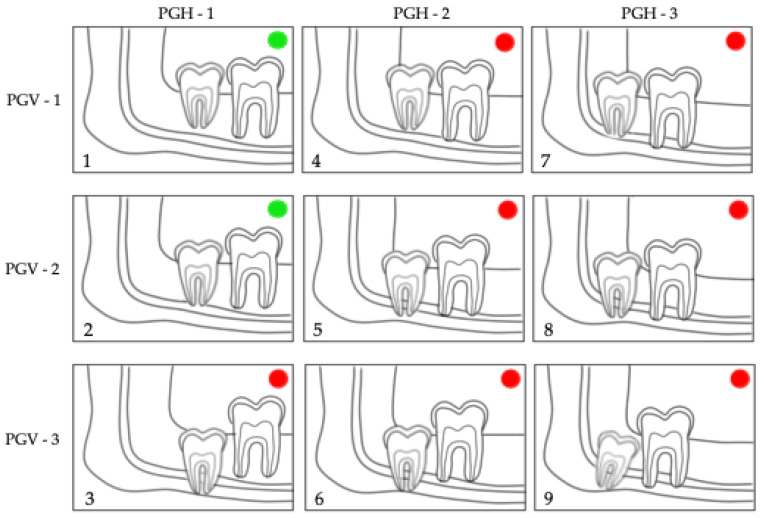
Pell & Gregory´s classification of mandibular M3 in relation to the horizontal (PGH) and vertical (PGV) position of M2 [25]. Reference plane/structure: the occlusal plane (PGV) and the ramus (PGH). Subdivisions according to the eruption prognosis of M3 in **1**,**2**: good (
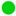
); **3–9**: bad prognosis (
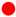
).

**Figure 6 jcm-12-01060-f006:**
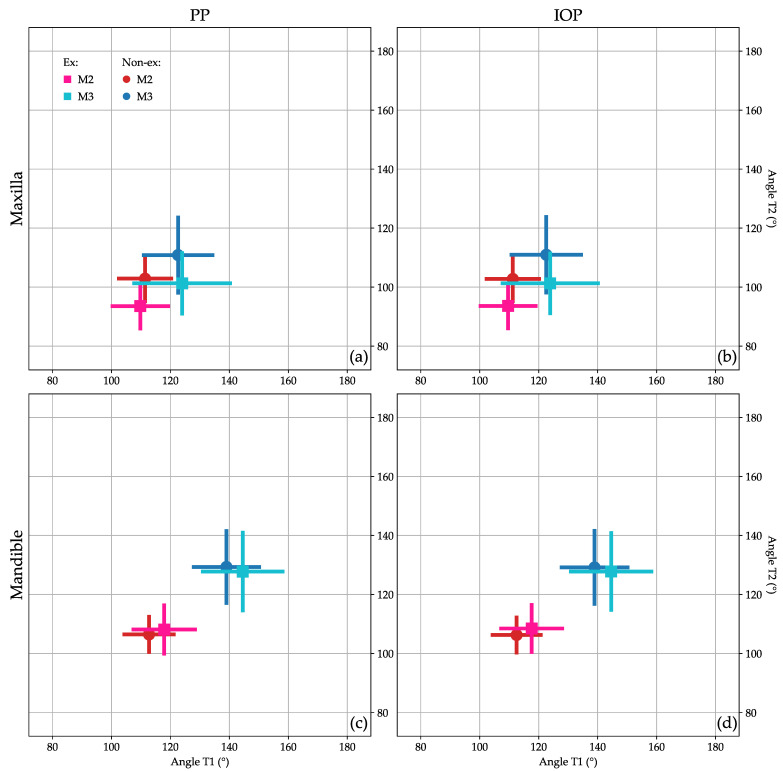
Patients´ angular changes of M2 and M3 between the two examination time points (T1: x-axis, T2: y-axis). Means, standard deviations (cross), and distribution of the non-extraction (Non-ex): (M2: 
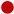
, M3: 
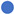
) and extraction (Ex): (M2: 
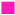
, M3: 
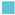
) groups. (**a**) maxillary M2/PP and M3/PP; (**b**) maxillary M3/IOP and M3/IOP; (**c**) mandibular M2/PP and M3/PP; (**d**) mandibular M2/IOP and M3/IOP.

**Figure 7 jcm-12-01060-f007:**
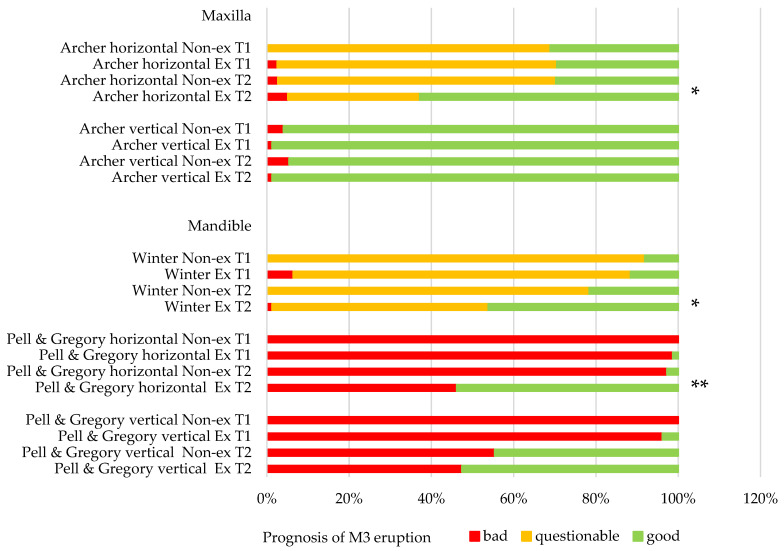
Distribution of the eruption prognosis of M3 in non-extraction (Non-ex) and extraction groups (Ex), before (T1) and after orthodontic space closure (T2) according to the different classification systems. * *p* < 0.05; ** *p* < 0.001.

**Table 1 jcm-12-01060-t001:** Patients’ distribution in extraction and non-extraction groups according to sex, age, treatment duration, Angle classification, skeletal classification, and growth pattern.

		Non-Extraction (*n* = 37)	Extraction (*n* = 39)
Sex (*n* (%))	female	23 (62.16)	24 (61.54)
	male	14 (37.84)	15 (38.46)
Age pretreatment (years)	mean	14.5	13.5
	range	9.6–19.4	11.8–15.3
Age post-treatment (years)	mean	16.2	16.2
	range	11.2–21.1	14.4–18.0
Treatment duration (years)	mean	1.65	2.99
Angle classification (*n* (%))	Class I	11 (29.73)	18 (46.15)
	Class II	22 (59.46)	13 (33.33)
	Class III	4 (10.81)	8 (20.52)
Skeletal classification (*n* (%))	Class I	10 (27.03)	15 (38.46)
	Class II	20 (54.05)	15 (38.46)
	Class III	7 (18.92)	9 (23.08)
Growth pattern (*n* (%))	neutral	18 (48.65)	14 (35.90)
	vertical	2 (5.40)	12 (30.77)
	horizontal	17 (45.95)	13 (33.33)

**Table 2 jcm-12-01060-t002:** Angular measurements in maxilla and mandible, adjusted treatment effects at T2, 95% confidence intervals, and *p*-values for the angular measurements per measurement and jaw.

		T1 Mean (sd)	T2 Mean (sd)	Coefficient (95% CI)	*p*-Value
** Maxilla **					
M2/IOP	Non-ex	111.2° (8.9)	102.8° (7.5)	Reference	
	Ex	109.6° (9.4)	93.6° (7.6)	−8.72 (−11.38, −6.07)	<0.001 **
M2/PP	Non-ex	111.4° (8.9)	102.9° (7.6)	Reference	
	Ex	109.8° (9.4)	93.6° (7.6)	−8.83 (−11.51, 6.15)	<0.001 **
M3/IOP	Non-ex	122.6° (11.6)	111.0° (12.8)	Reference	
	Ex	124.0° (16.3)	101.3° (10.1)	−10.17 (−14.21, 6.13)	<0.001 **
M3/PP	Non-ex	122.6° (11.6)	110.9° (12.8)	Reference	
	Ex	124.0° (16.3)	101.3° (10.2)	−10.04 (−14.14, 5.95)	<0.001 **
** Mandible **					
M2/IOP	Non-ex	112.5° (8.2)	106.2° (6.0)	Reference	
	Ex	117.7° (10.4)	108.5° (8.0)	0.73 (−1.65, 3.11)	0.55
M2/PP	Non-ex	112.7° (8.4)	106.4° (6.0)	Reference	
	Ex	117.9° (10.5)	108.1 (8.2)	0.72 (−2.17, 2.72)	0.83
M3/IOP	Non-ex	138.9° (11.2)	129.2° (12.4)	Reference	
	Ex	144.6° (13.6)	127.8° (13.0)	−4.03 (−8.23, 0.17)	0.06
M3/PP	Non-ex	139.0° (11.1)	129.3° (12.2)	Reference	
	Ex	144.5° (13.5)	127.7° (13.2)	−4.20 (−8.38, −0.02)	0.05

M1 = first molar; M2 = second molar; M3 = third molar; IOP = infraorbital plane; PP = palatal plane; Non-ex = non-extraction group; Ex = extraction group; T1 = before orthodontic treatment; T2 = after orthodontic space closure; sd = standard deviation; 95% CI = 95% confidence interval; ** *p* < 0.001.

**Table 3 jcm-12-01060-t003:** Odds ratios and 95% confidence intervals from the random effects analysis for the different classification systems (Archer, Winter, Pell & Gregory).

		Archer Horizontal	Archer Vertical	Winter	Pell & Gregory Horizontal	Pell & Gregory Vertical
		OR (95% CI)	OR (95% CI)	OR (95% CI)	OR (95% CI)	OR (95% CI)
Group	Non-ex (Reference)					
	Ex	1.96 *	21.79	1.92	2234.22 *	2.35
		(1.04, 3.72)	(0.17, 2786.99)	(0.40, 9.15)	(11.35, 439,962.1)	(0.62, 8.94)
Time	T1 (Reference)					
	T2	1.71	0.28	23.93 **	5459.58 *	509.77 **
		(0.91, 3.23)	(0.01, 10.21)	(3.56, 152.23)	(16.53, 1,802,824)	(39.66, 6522.06)
Sex	Female (Reference)					
	Male	0.66	0.55	4.16	6.55	2.54
		(0.34, 1.29)	(0.02, 15.94)	(0.78, 23.32)	(0.41, 105.77)	(0.65, 10.02)
Crowding	No crowding (Reference)					
	<3 mm	0.93	1.67	1.10	14.46	3.44
		(0.46, 1.85)	(0.04, 69.99)	(0.20, 6.14)	(0.61, 343.72)	(0.78, 15.26)
	3–5 mm	0.46	0.49	0.68	77.98	5.53
		(0.16, 1.32)	(0.01, 42.03)	(0.05, 8.95)	(0.78, 8482.21)	(0.68, 44.92)
	>5 mm	0.98	1.50 × 10^13^	0.004 *	0.24	24.32
		(0.14, 6.78)	(0, non-estimable)	(0.0002, 0.95)	(0.47, 186,379.3)	(0.40, 1475.36)

Non-ex = non-extraction group; Ex = extraction group; T1 = before orthodontic treatment; T2 = after orthodontic space closure; OR = odds ratio; 95% CI = 95% confidence interval; * *p* < 0.05; ** *p* < 0.001.

## Data Availability

The data presented in this study are available on reasonable request from the corresponding author.

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
