# Peer review of "Eruption Pattern of Third Molars in Orthodontic Patients Treated with First Permanent Molar Extraction: A Longitudinal Retrospective Evaluation"

_jcm, 2023, doi:10.3390/jcm12031060_

Round 1

Reviewer 1 Report

Reviewer Comments on “Eruption pattern of third molars in orthodontic patients treated with first permanent molar extraction. A longitudinal retrospective evaluation.”

Summary

This article aims to explore the effects of first molar extractions on second and third molar positioning. The study also evaluated the effect of first molar extraction on third molar eruption potential. The team retrospectively analyzed pre-treatment and post-treatment radiographs of patients who underwent extraction and non-extraction treatment using both quantitative angular measurements and qualitative categorization to evaluate the effects. The study concluded that first molar extraction treatment can have a positive effect on second and third molar positioning in the maxilla, but the results did not indicate a favorable effect on the mandibular molars.

Recommendation concerning publication: Major Revisions

This study is clinically relevant and would provide general dentists, pediatric dentists, orthodontists and oral surgeons with pertinent information when considering first molar extraction treatment. The study presents relevant clinical implications when extracting first molars and is well written. That being said, the size and heterogeneity of the study could be improved, and stratifying data by eruption status and developmental stage of the 2nd molars would also benefit the study. With revisions, I could see recommending this study for publication.

Major Revisions:

Abstract

-Overall well written abstract that conveys the aim, methods, and findings.

-You indicate here that T2 is after orthodontic treatment, but I believe T2 is specifically after completion of orthodontic space closure and not necessarily following completion of all orthodontic treatment. This might be slightly misleading considering the outcomes may be slightly different at a time point only after space closure vs. a time point after completion of orthodontic treatment

-It unclear what the difference is between angulation and inclination when reading the abstract. Please clarify.

-You mention the findings of the investigation as a favorable angulation change in the maxilla and no favorable angulation effect in the mandible. It is worth noting that the inclination improved in both the maxilla and the mandible as a significant finding of the investigation.  It seems somewhat inconsistent to mention the angulation, but not the inclination change that was significant for both max and man.

-Please clarify whether the eruption prognosis of third molars improved with first molar extraction.

Introduction

-Nicely written introduction, but quite brief. Please expand on the impacts of M3 and M2 impaction and costs/risks inherent in that. Please expand on prior work evaluating effects of M1 extraction on M2 and M3.

-You mention here that MIH has risen most dramatically in 6-14 year olds. Why has it increased?  Please clarify.

Methods:

-Eruption status or the development status of the second molars and third molars could have a big effect on positioning of the molars and how much they improve. You looked at Demirjian’s classification, but did you consider limiting the age range of patients or better yet, stratifying your data based on unerupted versus already erupted second molars at T1? This would help limit confounding and help to isolate variables. It would help to address the question of- Does M1 extraction only have an impact on distal molars (M2/M3) early in development or does it impact positioning at any time?

-How far prior to the start of T1 could the pre-treatment pan have been taken? Please specify in your paper. This could have an effect on the accuracy of your molar starting position, as time was an important predictor of outcomes.

-As with any study, having a larger, more diverse sample would improve your conclusions. However, you had significant data, and we are all bound by the limits of our data set.

-How reliable are your angular measurements? Could the right or left reference points having different height positions affect your angles? You noted excluding for asymmetries. How did you identify your asymmetries to be excluded?  Based on what criteria?

-Was the method of space closure recorded and factored into your analysis (reciprocal space closure or absolute with TADs)? From what you describe, it does not appear so. These aspects might affect the positioning of the molars after space closure.

-How did you classify your subgroups (Good, bad, questionable)? What is this based on? Is there any evidence to support the subgroup classifications and their likelihood of spontaneous M3 eruption? The original classifications were used more to predict difficulty of extraction or to help determine the path of extraction needed, as opposed to likelihood of spontaneous eruption.

Results and Discussion:

-Have you considered whether the bigger differences in starting points of the mandibular molar T1 angulations between Non-ex and ex affect your outcomes? Please address this..

-In my opinion, some of your most interesting results/findings for clinicians are noted between lines 215-220. It could be beneficial to highlight these implications in your discussion more.

-You noted in your discussion at line 252 that the severe mesial inclination of the mandibular molars could be effecting outcomes. If so, why not stratify your data based on molar inclination (mesial versus distal) to see how that could possibly affect the potential for positioning improvement?

- You point out in the discussion that : Former studies found that patients with a dolichofacial 295 growth pattern showed a twofold prevalence of mandibular M3 impaction than patients 296 with a horizontal growth pattern (brachyfacial) [17, 50, 51].  Is there any difference between malocclusion classification and impaction of molars.- Could you stratify your data by Class II, III or I to see if that had any effect on outcomes?

Tables and Figures

-In Table 1, please include the number of unilateral and bilateral extractions which you mention in methods.

-I like the idea in Figures 2-5 of subgrouping these categories by color based on potential eruption outcome. However, as noted above- Is there some reasoning behind how you identified eruption potential for each subgroup? Why do some of these classifications have good and bad, but some also have questionable? Could you have combined all these classification systems you utilized to create a simpler categorization system of your own and consequently communicate your findings more effectively?

-Figure 6 is a nice visual. That being said, this is a difficult set of graphs to follow initially because it is unclear what changes in angulation or shift in the graph indicate an improvement in position without deeper understanding of the study. Could this be presented in a way that makes it clearly evident what is being shown? Also, please clarify in the legend and possibly the visuals that the square are extractions and circles are non-ext. It took me a minute to figure that out.

Minor Revisions:

Page 1, Line 35: “over” may need to be added before “the recent years” or potentially replace “the”

Page 5, Line 168: “non-extractiom” change to “non-extraction”

Page 6, Line 181: Why is “permanent” present for M1, but not M2 or M3?

Page 9, Line 242-243: Consider removing “seldomly” or using “rarely”; This comes off as confusing

Page 9, Line 244: Consider “could” instead of “should”

Page 10, Line 289: Consider “lingually” as opposed to “orally”

Author Response

Dear Reviewer,

Reviewer 2 Report

This is basically very informative study to the clinicians. Article is well written, and its shortcomings are well addressed in the discussion section. Presented graphics are very intuitive and easy to understand. One thing I would like to recommend is adding the time duration between T1 and T2 for both the extraction group and control group.

Author Response

Response to Reviewer 2 Comments

Point 1: This is basically very informative study to the clinicians. Article is well written, and its shortcomings are well addressed in the discussion section. Presented graphics are very intuitive and easy to understand. One thing I would like to recommend is adding the time duration between T1 and T2 for both the extraction group and control group.

Response 1: Thank you very much for your positive comment and advice. The treatment duration for both groups is already mentioned in table 1 and in the text. “The orthodontic treatment time was significantly longer for patients treated with extraction of M1 than those treated without extraction (p < 0.001), as expected.” (page 5, lines 154-156).

But you are absolutely correct. We didn’t mention the time period between the pre- and posttreatment panoramic radiographs. Thanks for drawing our attention to it. We included it: “The period between the pre-and posttreatment radiographs was 3.8±1.3 years in the non-extraction group and 3.7±1.4 years in the extraction group.” (page 5, lines 156-158).

Reviewer 3 Report

Dear authors,

the article is very interesting. The topic is very up-to-date and could be interesting for the readers. Nevertheless, I suggest providing some modifications in order to improve the manuscript.

Lines 35-36.

The authors could also focus on the problems related to the consequences of the delays in dental treatments caused by the pandemic situation.

The authors could add the following sentence:

“MIH diagnosis occurs late, thus causing the need for more invasive treatments, such as extraction. This has been further aggravated by the pandemic situation which has caused delays in diagnosis and dental treatments”.

The reference https://doi.org/10.23736/S2724-6329.21.04632-5 could be used to support this sentence.

Lines 35-39:

In the introduction, the importance of early MIH diagnosis shall be outlined. Prognosis change completely (and therefore the need for M1 possible extraction) according to the MIH diagnosis stage.

Table 1:

The authors could consider adding unilateral or bilateral extraction data to the table.

Lines 239-246.

The reviewer humbly thinks that a sentence related to the importance of early diagnosis in MIH should be added.

Early diagnosis change completely the prognosis of MIH affected teeth. From these lines it may be speculated that extraction is the best option. The authors could outline this concept (importance of early diagnosis) supporting with: Garg N, Jain AK, Saha S, Singh J. Essentiality of early diagnosis of molar incisor hypomineralization in children and review of its clinical presentation, etiology and management. Int J Clin Pediatr Dent. 2012 Sep;5(3):190-6. doi: 10.5005/jp-journals-10005-1164. Epub 2012 Dec 5. PMID: 25206166; PMCID: PMC4155885.

The authors could add a sentence/paragraph on possible differences and consequences on orthodontic/occlusal situation between unilateral or bilateral M1 extraction treatments.

Author Response

Response to Reviewer 3 Comments

Point 1: Dear authors, the article is very interesting. The topic is very up-to-date and could be interesting for the readers. Nevertheless, I suggest providing some modifications in order to improve the manuscript.

Response 1: Thanks for your positive comment and suggested modifications to improve our manuscript.

Point 2: Lines 35-36.

The authors could also focus on the problems related to the consequences of the delays in dental treatments caused by the pandemic situation.

The authors could add the following sentence:

“MIH diagnosis occurs late, thus causing the need for more invasive treatments, such as extraction. This has been further aggravated by the pandemic situation which has caused delays in diagnosis and dental treatments”.

The reference https://doi.org/10.23736/S2724-6329.21.04632-5 could be used to support this sentence.

Response 2: Thanks for your advice. We included your well-considered suggestion to add possible effects of a delayed MIH diagnosis. As the pandemic situation did not exist during orthodontic treatment of the study`s included patients, we did not address this point so far. Nonetheless, it is potentially relevant at this point. We included your sentence in the introduction (page 1, lines 38-41).

Point 3: Lines 35-39.

In the introduction, the importance of early MIH diagnosis shall be outlined. Prognosis change completely (and therefore the need for M1 possible extraction) according to the MIH diagnosis stage.

Response 3: We included your suggestion in the manuscript (page 1, lines 38-41).

Point 4: Table 1:

The authors could consider adding unilateral or bilateral extraction data to the table.

Response 4: We considered this point during preparation of the manuscript. Some patients were treated with bilateral extraction in one jaw but unilateral in the other jaw. For reasons of clarity, we have therefore decided against including it in table 1. 

Point 5: Lines 239-246.

The reviewer humbly thinks that a sentence related to the importance of early diagnosis in MIH should be added.

Early diagnosis change completely the prognosis of MIH affected teeth. From these lines it may be speculated that extraction is the best option. The authors could outline this concept (importance of early diagnosis) supporting with: Garg N, Jain AK, Saha S, Singh J. Essentiality of early diagnosis of molar incisor hypomineralization in children and review of its clinical presentation, etiology and management. Int J Clin Pediatr Dent. 2012 Sep;5(3):190-6. doi: 10.5005/jp-journals-10005-1164. Epub 2012 Dec 5. PMID: 25206166; PMCID: PMC4155885.

Response 5: Thanks for your recommendation. The decision to extract teeth affected with MIH should be encountered carefully. We added the importance of an early diagnosis to allow preventive measures (page 10, lines 266-269).

Point 6: The authors could add a sentence/paragraph on possible differences and consequences on orthodontic/occlusal situation between unilateral or bilateral M1 extraction treatments.

Response 6: Thanks for your suggestion. We did include this into our discussion as well. “Compensation extraction on the quadrants with sound molars is recommended for symmetric dental arches and balanced occlusion [39]. Unilateral extraction might require higher orthodontic anchorage to avoid a dental midline shift towards the extraction side [40] but has less impact on soft tissue and facial profile [41].” (page 11, lines 293-296).

Round 2

Reviewer 1 Report

Improved manuscript with greater clarity. I recommend it for publication.